# Antimicrobial Susceptibility of Microbiota in Bacterial Vaginosis Using Fluorescence In Situ Hybridization

**DOI:** 10.3390/pathogens11040456

**Published:** 2022-04-11

**Authors:** Alexander Swidsinski, Alexander Guschin, Lorenzo Corsini, Vera Loening-Baucke, Lenka Podpera Tisakova, Sonja Swidsinski, Jack D. Sobel, Yvonne Dörffel

**Affiliations:** 1Molecular-Genetic Laboratory for Polymicrobial Infections and Biofilms, Charité CCM, Medizinische Klinik, Universitätsmedizin, 10117 Berlin, Germany; veraloening@gmail.com; 2Institute of Molecular Medicine, Sechenov First Moscow State Medical University, 119435 Moscow, Russia; 3Moscow Scientific and Practical Center of Dermatovenerology and Cosmetology, 119071 Moscow, Russia; alegus65@mail.ru; 4BioNTech R&D (Austria) GmbH, Vienna Biocenter, 1110 Vienna, Austria; lorenzo.corsini@phagomed.com (L.C.); lenka.tisakova@phagomed.com (L.P.T.); 5MDI Limbach GmbH, Aroser Allee 84, 13407 Berlin, Germany; sonja.swidsinski@mvz-labor-berlin.de; 6School of Medicine, Wayne State University, Detroit, MI 48201, USA; jsobel@med.wayne.edu; 7Outpatient Clinic, Charité Universitätsmedizin Berlin, CCM, 10117 Berlin, Germany; yvonne.doerffel@charite.de

**Keywords:** antimicrobial resistance, microbiome susceptibility, bacterial vaginosis, dysbiosis, FISH, polymicrobials, metronidazole, octenisept®, ciclopirox, *Lactobacillus crispatus*, *Saccharomyces boulardii*, *Gardnerella*-phage-endolysin, phagolysin

## Abstract

Background: Testing of antibiotic resistance of intact vaginal microbiota in pure culture is not feasible. METHODS: Metronidazole, antiseptic octenisept^®^, antimycotic ciclopirox, bacterial probiotic *Lactobacillus crispatus*, yeast probiotic *Saccharomyces boulardii, Gardnerella*-phage-endolysin named phagolysin and phagolysin in combination with probiotics were tested for bacteriolytic activity. Included were vaginal swabs from 38 random women with Amsel-confirmed bacterial vaginosis (BV). Test aliquots were incubated by 37° for 2 and 24 h. *Gardnerella*, low G+C, *Atopobium*, lactobacilli, *Lactobacillus iners* and *crispatus*, *Prevotella*-*Bacteroides, and Gammaproteobacteria* microbial groups were quantified using fluorescence in situ hybridization (FISH). Results: The probiotic strain *Lactobacillus crispatus* demonstrated the weakest bacteriolytical effects, followed by metronidazole. Both had no impact on *Gardnerella* species, instead lysing *Prevotella-Bacteroides*, *Enterobacteriaceae* (by *L*.*crispatus*) or LGC, *Atopobium* and *Prevotella-Bacteroides* (by metronidazole) groups of the microbiota. Cytolytic activity on *Gardnerella* was highly pronounced and increased from octenisept to ciclopirox, phagolysin, phagolysin with *L.crispatus,* being best in the combination of phagolysin with *S*.*boulardii*. Universally active ciclopirox and octenisept® suppressed nearly all microbial groups including those which are regarded as beneficial. Phagolysin had no effect on naturally occurring *Lactobacillus crispatus.* Conclusions: FISH susceptibility testing allows unique efficacy evaluation of individually adjusted topical therapy without microbial isolation facilitating optimal therapy choice.

## 1. Introduction

Bacterial vaginosis (BV) is the most common cause of vaginal discharge [1]. In addition to troublesome symptoms, BV is associated with serious complications including preterm birth, infertility and higher STI risk, among others [1,2].

The disease is characterized by a marked increase in bacterial diversity and concentrations [1,2]. Although involved microorganisms are presently identified by next generation sequencing, exact details of pathogenesis are still controversial [3]. *Gardnerella* species are omnipresent and dominant in nearly all cases, but are also often present in healthy women [3,4]. Occurrence of other pathogenic anaerobes, even when found exclusively in BV, is unpredictable and variable. BV is therefore generally defined as a disappearance of the normally lactobacilli-dominated vaginal microbiota overgrown and replaced by gram-labile bacteria of which *Gardnerella* species are most numerous, forming structured biofilms [3,4]. The recognition of such a microbial shift is already apparent on wet microscopy, but the polymicrobial nature represents a therapeutic challenge [3]. Traditionally, the central tool of infectious disease research is susceptibility testing and measuring growth-inhibition of pure cultures of individual microbial strains. However, cultured isolates respond to antimicrobials differently than microorganisms which are growing in the form of a polymicrobial community.

Current treatment principles of vaginal dysbiosis are therefore based on estimations derived from in vitro susceptibility tests of individual isolates, clinical observations and experience based upon group statistics.

Theoretical models and personal preferences led to a long list of drugs and measures with general low efficacy and high recurrence rates [5]. Susceptibility tests of native polymicrobial communities, which are necessary for evidence-based therapy are lacking. Although modern gene-based detection methods allow characterization of growth/suppression dynamics of polymicrobials, their application for susceptibility testing has not emerged. Two problems interfered. First, as opposed to isolated strains, the growth of complex microbiota outside of their natural environment is unpredictable. Each utilized culture medium favors single components of the complex microbiota differently making a balanced growth of the involved polymicrobials microbiome even for the short time in resistance testing impossible. Irrelevant bystanders overgrow and suppress constitutive microorganisms in vitro with or without antimicrobial additives. Second, it is difficult to distinguish between living organisms and their DNA remnants by culture independent methods. Both difficulties can be circumvented.

Copan Liquid Amies Elution Swab (ESwab™ 493C02 COPAN) Collection and Transport System contain puffer without nutritive additives. Vaginal smears obtained from the vaginal surface include microbiota, copious slime and desquamated epithelial cells, on which the microbiota were previously growing. The transfer in eSwab maintain the pre-existing conditions exactly the same as in the individual patient from whom they came, except for the dilution effect. Test substances can be incubated with aliquots of such suspensions providing reliable reference to each other and controls.

Ribosomal-gene-based in situ hybridization (FISH) targets intact microorganisms at species, group and domain level. Presence of microbial DNA in samples is irrelevant, since FISH signals must correlate with definite morphology to be distinguished from biases. Although FISH cannot discriminate between living and dead microorganisms (microbes are fixed at the time of FISH), the visual disappearance of the previously documented intact microbial cells indicates, that they were lysed by test substances. In the present study, we quantified the bacteriolytic effects of six common antimicrobials reported to have clinical efficacy and representing different modes of action.

## 2. Patients, Materials and Methods

Samples from 40 randomly selected 21–48-year-old (mean 29 years) white symptomatic women with BV diagnosed according to the Amsel criteria. Presence of sexually transmitted disease was excluded. None of the women had received an antibiotic in the previous 6 months.

ESwab™ 493C02 COPAN Diagnostics were used for collection of vaginal smears and stored at 4 °C Celsius for not longer than 24 h prior to resistance testing.

### 2.1. Antimicrobial Susceptibility Test

To avoid uneven distribution on the microscopic slide of heterogeneous suspensions of vaginal smears, quantifications of bacteria need a reliable reference. The best reference provides a definite number of epithelial cells with surrounding and attached bacteria, since this is independent of dispersion biases. The sufficient number and maintenance of the epithelial cells over time of testing is therefore crucial for measurement validity.

In total, 10 µL of suspensions from vaginal smears taken in Copan E-swabs contained upon microscopy at least 50 epithelial cells within a 10 mm × 10 mm area at magnification of 400. In most cases however, more than 1000 epithelial cells were observed, comfortably allowing reliable quantification of bacteria related to epithelial cells and minimizing possible dilution and manipulation biases. Dilution by addition of antimicrobial test substances, incubation, Carnoy-fixation and resuspension steps led to about 30% loss of the initial number of epithelial cells, which needed to be compensated for by the reduced volume in which the samples were dissolved after all procedural steps. Based on this algorithm, 1000 µL of Amies medium in Copan eSwab provided sufficient material for 9 different antimicrobial tests (100 µL of the suspension + 50 µL of the test substance) for 2 h and 24 h incubation (75 µL of the mix for each) still leaving material (left directly on the swab) for other cultural, PCR and sequencing based techniques when necessary.

The incubated 100 µL of the vaginal suspension with 50 µL of test substance, terminated by Carnoy, centrifuged and resuspended in volume of 70 µL were in turn sufficient for about 10 multicolor hybridizations for each set.

### 2.2. Susceptibility Test

The filled eSwab-vials were vortexed gently, resulting in a suspension that contained epithelial cells and vaginal bacteria. The suspensions were divided in 10 portions of 100 µL each. All reactions and the following steps were performed in Eppendorf 1.5 mL vessels.

In total, 50 µL of the following test substances or their combinations were added dissolved in required puffer/medium to achieve the following concentrations of additives: none for untreated controls, metronidazole (1 mg/mL), ciclopirox (0.5 mg/mL), octenisept® vaginal (dilution 1:20), *Lactobacillus crispatus* (10^8^ cfu/mL suspension), *Saccharomyces boulardii* (10^7^ cfu/mL suspension), phagolysin (20 µg/mL), phagolysin + *Lactobacillus crispatus*, phagolysin + *Saccharomyces boulardii*. 50 µL Amies medium were added to untreated controls or to tests of antimicrobials to equalize the dilution.

Metronidazole (B. Braun Melsungen AG, Germany), ciclopirox (Almirall Hermal GmbH, Reinbek, Germany), octenisept® (Schülke, Norderstedt, Germany) and *Saccharomyces boulardii* CNCM I-745 (Biocodex, Gentilly, France) were purchased from the pharmacy. *Lactobacillus crispatus* was obtained from Leibnitz Institute DSMZ German Collection of Microorganisms and Cell Cultures GmbH, Phagolysin was provided by PhagoMed Biopharma GmbH. For this study engineered *Gardnerella* phage endolysin PG100 was used. For convenience reasons we refer to this preparation as “phagolysin” throughout the manuscript. In comparative investigations of lytic activity, phagolysin proved to be 100-fold more effective (data not shown) than the previously extensively tested phage endolysin pH477. We could therefore use lower concentrations than in the previous study [6].

The 150 µL of test aliquots were incubated aerobically at 37 °C and divided. After 2 and also 24 h, the action was terminated by adding 1000 µL Carnoy solution (alcohol/chloroform/acetic acid 6/3/1 by volume) [7] for 10 min. The samples were than centrifuged at 6000 RPM for 8 min, and taken and stored in 50 µL of Carnoy solution.

### 2.3. FISH

All identifications and quantifications of microorganisms in this study were performed exclusively based on FISH methodology.

Fields of 10 mm × 10 mm were marked on SuperFrost slides (Langenbrinck, Emmendingen, Germany) with a PAP pen (Kisker-Biotech, Steinfurt, Germany). Five μL aliquots of vortexed fixed antimicrobial resistance tests were dropped onto the marked field. The slides were dried for 60 min at 50 °C before FISH analysis.

Principally, any of the microbial groups composing the biofilms can be quantified. However, the number of hybridizations performable with a single sample are limited. The main focus of the present investigation was on populations of *Gardnerella species*, *Atopobium*, low GC = LGC (guanine + cytosine) bacteria (*Mycoplasmatales*, *Firmicutes*, *Bacillales*, *Lactobacillales*), lactobacilli, *Lactobacillus iners*, *Lactobacillus crispatus*, *Prevotella-Bacteroides*, *Gammaproteobacteria* and *Candida species*. We do not know which of these groups if any are crucial for dysbiosis. However, *Gardnerella* and *LGC* bacteria were especially suitable to quantify the antimicrobial effects since they were most numerous and ubiquitous within vaginal dysbiotic communitiy and their FISH signals do not overlap, which was especially valuable for the quantitative assessment.

The following probes were applied: Gard662 (*Gardnerella*), Lab158 (lactobacilli) Liner23-2 (*L.iners),* Lcrips16-1 (*L.crispatus*), Ato291 (*Atopobium*), Caal (*Candida*) [8], low G+C probe LGC35 [9], Bact1080 [10] (*Prevotella-Bacteroides*), GAM42a (*Gammaproteobacteria*) and in special cases also Eub338 (all bacteria) [11] and Ebac1790 (*Enterobacteriaceae*) [12].

For multicolor analysis each the applied oligonucleotide probes were synthesized with a carbocyanine Cy3 (orange) and Cy5 (dark red) fluorescent dye. The hybridizations were performed at 50 °C as previously described [7]. DAPI stain was used to visualize DNA rich structures of bacteria and eukaryotic cells. In some cases, we applied dobe-FITC stained Eub338 probes to see all bacteria in the same set in addition to DAPI.

Hybridizations were performed with at least two differently stained probes. The Cy3 stained probe served the enumeration of the targeted bacterial cluster, Cy5 stained probe served as reference to other microbiome groups. We used Gard662-C3/LGC-C5 initially and combinations of C3 stained other microbial probes with Gard662-C5 probe as a universal reference. In the presence of high density or local microbial accumulations, the differentiation of single microorganism became sometimes difficult using the universal for all bacteria Eub338 probe as reference, because it hybridizes with exactly the same microorganisms as the FISH probe of interest.

A Nikon e600 fluorescence microscope was used. The images were photo-documented with a Nikon DXM 1200F color camera and software (Nikon, Tokyo, Japan).

The exact protocols with single steps and solutions are also presented at: http://www.swidsinski.de/zusatzdateien/fishmethode/fishmethode.htm(accessed on 1 April 2022).

### 2.4. Bacterial Quantification

Bacteria were enumerated in a representative area surrounding 10 epithelial cells. When low numbers of bacteria were present (for example lactobacilli, *Prevotella*, *Gammaproteobacteria*, etc.) larger areas, including at least 10 microscopic fields were evaluated and the mean number referred to 10 epithelial cells was used. When moderate numbers with spaces between bacteria were present, all countable bacteria were enumerated. In case of too high local concentrations within biofilms with no spaces between individual bacteria the counting was impossible. In such cases, the 10 × 10 µm surface of such agglomerates was assigned to 500 bacteria. The total surface covered with such biofilm was measured in area around 10 representative epithelial cells, expressed in numbers of full quadrats, multiplied with 500 and rounded to 100. Individually countable numbers of bacteria within areas of 10 epithelial cells were added to biofilm derived counts when exceeding 100.

Such mode of enumeration makes a direct comparison between moderate and high-density samples questionable. To reduce the possibility of quantification bias, we subdivided the patients in two subgroups with moderate (<500 bacteria, N = 22) and high density (>500 to 5000 bacteria, N = 18) dysbiosis and evaluated them separately. This graduation in high and moderate density was performed based on enumeration of bacteria seen in the DAPI stain.

### 2.5. Exceptions and Anomalies

An excessive growth of *Candida* completely destroyed epithelial cells after 24 h of incubation, making the comparable quantification impossible in two of the patients with moderate density dysbiosis. These two samples/patients were removed from analysis.

The incubation with phagolysin for 2 h at 37 °C led to a decrease in *Gardnerella* numbers (mean ± SD 840 ± 920), while LGC bacteria remained unchanged (mean ± SD 161 ± 139). In the 2h sample from the control group the decreased values were (mean ± SD 1380 ± 1371 and 171 ± 168). No relevant changes in the mean numbers of *Gardnerella* or LGC bacteria were observed with all other antimicrobials as compared to the untreated controls after 2 h incubation (data of other test groups are not shown for irrelevance). No further hybridization or analysis of samples incubated for 2 h were performed.

The differentiated response of bacteria to other than phagolysin antimicrobials became obvious after 24 h at 37 °C of incubation. These samples were hybridized with FISH probes for all chosen microbial groups.

### 2.6. Use of Term Lysis

Within the frame of “classical antibiotic resistance concepts” the lytic action of antibiotics is used to describe direct degrading effects of the antimicrobial substances on the microorganisms. The direct impact of antimicrobials on bacterial cells was not und could not be demonstrated in our study, which monitors the numbers of physically present bacterial cells at initial, 2 h, and 24 h timepoints. We cannot say what exactly happens within these periods of time with the bacteria. Are bacteria directly disintegrated upon impact of antibiotics, are they weakened, dying or being decayed by other factors? What we actually measure with FISH are numeric dynamics of specific microbial groups within a polymicrobial population-rise of some and disappearance of others. 

However, the scientific literature uses the word “lysis” not only as in the above-described content but also in context of dissolvement, disappearance, solving, termination, etc. One should bear in mind, that speaking about bacteriolytic effects in our manuscript we do not mean direct disintegration of bacteria upon antimicrobial impact, but their disappearance following exposure to the test antimicrobials.

## 3. Results

General dynamics in vaginal microbiome upon incubation without therapeutic additives.

*Gardnerella* and LGC bacteria were present in all baseline samples. The baseline microbial concentrations of *Gardnerella* were mean ± SD 1494 ± 1474 for all, 2576 ± 1609 for high density dysbiosis and 156 ± 110 for moderate density dysbiosis. Other microbial groups were detected only in some samples, with lactobacilli (Lab 158), *Atopobium* (Ato291), *Prevotella* (Bacto1080) being most often (Figure 1, Table 1 and Table 2).

The mean numbers of *Gardnerella* species were nearly a Log_10_ higher than that of all other studied microorganism. Gram-positive *Gardnerella* composed 87%, *Atopobium* 9%, LGC 5%, and Gram-negative *Prevotella* + *Gammaproteobacteria* together 2% of all investigated vaginal bacteria in high density microbiome. In moderate density dysbiosis the ratio of *Gardnerella* numbers was lower with 61% (Gard), 8% (Ato), 22% (LGC) 9% (Gram-negative). The low GC bacteria are a large group including *Mycoplasmatales*, *Firmicutes*, *Bacillales*, *Lactobacillales*. The increase of its ratio in moderate density dysbiosis is probably due to the higher occurrence and concentrations of *Lactobacillus iners*.

All investigated microbial groups tended to fall upon incubation at 37 °C in similar rates leaving in each of the individual vaginal smears enough bacteria for comparison of antimicrobial effects. The proportion of bacterial groups to each other remained approximately the same in 28 of 38 samples with 24h incubation (exact relation for individual data not shown). The only changes observed were due to excessive increase (one to two < Log_10_) of *Candida* (4 of 8 were positive for *Candida samples*) or *Gammaproteobacteria* (6 of 18) with resulting unproportioned suppression of most other bacterial groups. This constancy of the individual microbiome profiles beyond the overgrowth exceptions was astonishing, since the vaginal fluid suspensions were incubated at aerobic conditions and the vaginal microbiome is composed of aerobe and oxygen sensitive anaerobes. Obviously, the polymicrobial environment in native vaginal slime can sufficiently sustain its components. The group specific overgrowth in some samples indicates that deprived of actively present vaginal immunity, some strains within vaginal microbiome escape the restraints of the previously protective community environment. The additional hybridization of samples with *Gammaproteobacteria* overgrowth demonstrated, that the increment is solely due to *Enterobacteriaceae* (Ebac1790 probe) probably representing some recent incomer. In two samples *Candida* overgrowth was thus massive that all epithelial cells were destroyed and had to be removed from the resistance evaluation. In four, the growth did not visually affect the epithelial cell numbers. These four *Candida* positive and six samples with *Enterobacteriaceae* overgrowth were not removed from the antibiotic resistance testing, since despite chaotic changes in ratios of the initially observed microbial group, each sample sustained enough microbial representatives for a comparison of antimicrobial effects.

## 3.1. Measurement of Antibiotic Resistance

All tested antimicrobials demonstrated significant lytic effects on at least one of the microbial groups, and the overall lytic patterns were agent specific and differed strongly from each other.

## 3.2. Gardnerella and LGC

While some of the antimicrobials led to dramatic changes in *Gardnerella* and LGC concentrations, distributions and form were hardly or not distinguishable from the controls (Figure 1 and Figure 2; Table 1 and Table 2).

Generally, the 24 h incubation led to a reduction in *Gardnerella* species and an increase in LGC bacteria, however the changes compared to baseline remained statistically insignificant *p* = 0.2 (Table 1). Statistically insignificant were the bacteriolytic impact of metronidazole, *Lactobacillus crispatus* or *Saccharomyces boulardii* although the mean concentrations of *Gardnerella* were 6%, 9%, and 26% lower in these groups than in the non-treated controls.

Most profound lytic effects (*p* < 0.001) were observed with ciclopirox, octenisept® and phagolysin, phagolysin combined with *Lactobacillus crispatus,* and *Saccharomyces boulardii*). Ciclopirox and octenisept® decimated both *Gardnerella* and LGC bacteria. No effect on LGC bacteria was observed with phagolysin, and an increase of LGC bacteria in phagolysin + *Saccharomyces boulardii* groups (*p* < 0.05).

Dividing the patients into those with moderate and high density dysbiosis (according to enumeration procedure described in the method section) did not change the observed effects of antimicrobials on *Gardnerella* and LGC bacteria for all but phagolysin + *Saccharomyces boulardii* group, where the previously observed tendency to lower *Gardnerella* numbers after exposing to probiotics became clearly statistically significant (*p* < 0.001), nearly completely eradicating *Gardnerella* in more than 75% of the samples.

### 3.3. 24 h Susceptibility of Other Microbial Groups

Measurable quantities of the *Atopobium*, lactobacilli, *Lactobacillus iners*, *Lactobacillus crispatus*, *Prevotella-Bacteroides*, *Gammaproteobacteria* and *Candida* groups were detected in 13% to 64% of the baseline vaginal samples and were still detectable in all corresponding control samples incubated for 24 h. Despite the low number of patients within single groups many of the antimicrobial effects reached a level of statistical significance (Table 2).

### 3.4. Atopobium

Octenisept® and ciclopirox significantly reduced *Atopobium* in all subgroups (*p* < 0.05–0.001). The effects of other antimicrobials were less impressive and often not clear in context of direct or mediated effects.

*Atopobium* numbers fell already with incubation of controls and mean numbers in all other test groups were lower than in controls reaching statistical significance (*p* < 0.05) only for phagolysin and phagolysin in combination with *L. crispatus* and *Saccharomyces boulardii* groups in moderate density dysbiosis. However, the parallelism of *Atopobium* and *Gardnerella* changes together with general instability of this microbial group indicates that the suppressive effects may be not due to direct antimicrobial action but be partially mediated by concomitant changes in *Gardnerella* availability.

### 3.5. Lactobacillus Species

Bacteria positive for Lab158 probe were sensitive to octenisept® and ciclopirox especially in the subgroup with high density dysbiosis. All other antimicrobials had no visible effects on them. *Lactobacillus iners* was sensitive towards *Saccharomyces boulardii* alone (ns) and in combination with phagolysin (*p* < 0.05).

*Lactobacillus crispatus* was detected in samples from three patients with high and two patients with moderate density dysbiosis. Only octenisept® and ciclopirox demonstrated direct effects on *L.iners* and *crispatus*.

### 3.6. Prevotella

Bacteria of the *Prevotella-Bacteroides* group were sensitive against metronidazole, octenisept®, intermediate to ciclopirox and *Lactobacillus crispatus* in combination with phagolysin.

### 3.7. Gammaproteobacteria

Different to all other investigated bacterial groups, the numbers of *Gammaproteobacteria* tended to be significantly (*p* < 0.05) increased upon incubation even in the controls. The increase was by Log_10_2 in two patients with high density dysbiosis and four patients with moderate density dysbiosis. The increase was clearly inhibited by octenisept®, ciclopirox and *Lactobacillus crispatus* alone and in combination with phagolysin, however the effects were not statistically significant for most because of the low number of patients (*p* = 0.07/0.08). Octenisept® appeared to be the only substance which significantly reduced *Gammaproteobacteria* (*p* < 0.05).

### 3.8. Candida

Similar to *Enterobacteriaceae*, *Candida* increased by more than one Log_10_ in four out of six included and by more than Log_10_2 in two patients, which had to be excluded from the investigation for excessive overgrowth. Only ciclopirox, octenisept® and *Sachharomyces boulardi* had a marked cytolytic, or growth suppressive effect on *Candida* (*p* < 0.01).

## 4. Discussion

Our data demonstrate that the differential lytic effects of antibiotics on complex microbial dysbiosis can be clearly quantified by FISH for each of the agents tested.

Interestingly, all of the chosen agents had a statistically significant lytic effect on at least one of the studied microbial groups. As expected, the single components of microbiota responded disparately to different antimicrobials with some of the groups being decimated, while other promoted. The effect varied in patients with high and moderate density dysbiosis. However, all observed effects in this study differed from those which we had anticipate based on our previous clinical practice and experience.

The mean values of multiple subgroups are difficult to compare at glance. To get a better overview using typical for clinicians susceptibility grading, in Table 3, (additional information only) we have sorted the effects presented in Figure 1 in the following steps: ±25% changed (R —resistant), decreased between 25%≥ and ≤Log_10_ (I—intermediately susceptible), decreased between Log_10_1≥ and ≤Log_10_2 (S—sensitive), decreased by more than ≥Log_10_2 (SS—highly sensitive), or increased accordingly 25%≥ <Log_10_1 (P—promoting).

Absolutely surprising was the complete indifference of *Gardnerella* species to metronidazole, the most often used antibiotic. The strong effects of metronidazole on Gram negative *Prevotella-Bacteroides* group were expected. However, both occurrence and numerical ratio of these bacteria on the whole dysbiosis was marginal. Unexpectedly, metronidazole displayed also intermediate effects on LGC bacteria in all and *Atopobium* in moderate density dysbiosis subgroup. However, the LGC group is very broad including *Mycoplasmatales, Firmicutes, Bacillales, Lactobacillales* [9] and the anaerobe *Firmicutes* species are sensitive to metronidazole.

As expected, metronidazole was the only of the tested substances, which promoted the growth of *Candida* (*p* < 0.05).

Likewise, highly impressive was the pronounced effect of the antifungal ciclopirox on all investigated participants of the vaginal microbiome regardless of bacterial density. The effect of ciclopirox on *Gardnerella* clearly exceeded that of octenisept® and was achieved only by phagolysin and exceeded that of the phagolysin + probiotics combination. Although ciclopirox is known to be suppressive for Gram-positive bacteria, it was not considered appropriate for antibacterial therapy. Ciclopirox was, as expected, best in the lysis of *Candida* cells.

The antimicrobial effects of octenisept® were in turn predictable and generally in accordance with all what we presently know. Compared with phagolysin and ciclopirox, octenisept® was less effective against *Gardnerella* (especially in high density dysbiosis) and *Candida* but exceeded effects of ciclopirox on *Atopobium*, lactobacilli (including *Lactobacillus iners* and *crispatus*), *Prevotella*-*Bacteroides* and *Gammaproteobacteria* groups.

The antimicrobial efficacy of probiotic strains was relatively low, when used alone. *Lactobacillus crispatus* had intermediate effects only on *Prevotella-Bacteroides* and *Gammaproteobacteria* (statistically not significant). *Saccharomyces boulardii* demonstrated intermediate effect on *Gardnerella*, on *Atopobium* (in patients with moderate dysbiosis density), on lactobacilli (in high density dysbiosis) and specifically on *Lactobacillus iners* in both dysbiosis groups, but only on *Candida* did the effects of *Saccharomyces boulardii* reach the level of statistical significance despite a very low number of the patients in this group. We are aware that in clinical practice, the probiotic *Lactobacillus species* were often used as combination of strains and were judged solely on evaluation of symptoms. However, *Lactobacillus crispatus* is regarded as the most potent, and none of the other components of such combinations were previously tested, while our data clearly indicate the necessity of such tests. On the other hand, probiotic microorganisms applied together with the biologic phagolysin potentiated the effect of phagolysin (or vice versa) on *Gardnerella,* especially in patients with moderate density dysbiosis leading to Log_10_2 reductions of *Gardnerella* after addition of *Saccharomyces boulardii.* Phagolysin combined with *Lactobacillus crispatus* had also intermediate effect on *Atopobium* in high density dysbiosis group (*p* < 0.05), on *Prevotella-Bacteroides* (*p* < 0.05), on *Gammaproteobacteria* (ns). Phagolysin in combination with *Saccharomyces boulardii* suppressed *Gardnerella* more intensively, lysed *Atopobium* in moderate density dysbiosis (ns), *L iners* (*p* < 0.05) and *Candida* in all dysbiosis groups (ns), displaying effects which were not observed by application of phagolysin alone.

The clear potentiating effects of combined applied biological agents demonstrate that in treating microbial communities scientifical data driven combination therapy may be preferable to achieve optimal clinical effects. Depending on initial composition of the microbiome and the individual aims, these combinations may include different antimicrobials, biologics, immune modulating substance and more.

## 5. Concluding Comments and Limitations

Although single forms of bacterial vaginosis are presumably sustained by a limited number of essentially responsible species, multiple accidently associated and transient microorganisms including bacteria, fungi, and phages make true relations difficult to untangle. The pathways by which polymicrobials prevail are poorly understood [3]. The present therapy of bacterial vaginosis is therefore highly problematic, purely empiric and employs a broad range of antibiotics, antiseptics, probiotics, and recently phage-derivatives [5,6].

In spite of the unclear etiopathogenesis, testing of antimicrobial susceptibility of the entire dysbiotic population could be crucial for effective therapy. This was impossible previously with methods based on isolation of single microorganisms. The development of FISH enabled direct monitoring of changes within complex microbial populations. We are aware of the fact, that chosen for this study marker-microorganisms, FISH probes and test conditions are far from optimal for the adequate assessment of clinically relevant antibiotic resistance. However, the classic culture based antibiotic susceptibility testing of isolated strains similarly needed years and many studies before it took its place in clinical practice. Antibiotic susceptibility of polymicrobials may similarly take many years.

Critically seen, the lytic activity measured may not be the same as killing activity, “not lysed” bacteria may be still affected by antimicrobials and non-vital. We are also aware that the ranges of lytic effects observed in vitro may strongly differ from those in clinical application. However, the same is true for any antimicrobial testing, and the in vitro determined minimal inhibitory concentration of classic tests have to be adjusted by corresponding clinical efficacy for each individual antimicrobial substance, regardless of effects observed in vitro. However, this new possibility of testing the efficacy of single agents alone and in combination for an individual dysbiotic microbiome opens compelling horizons in the treatment of challenging polymicrobial diseases such as BV and may prove to be a springboard for managing of all other polymicrobial diseases.

We acknowledge that this report does not answer all questions involved in this new methodology. However, this is the first effort made to approach susceptibility of a composite complex community of bacterial rather than each individual species involved. Additional studies are needed. This report serves as the first introductory study addressing the problem and supports several clinical observations.

## Figures and Tables

**Figure 1 pathogens-11-00456-f001:**
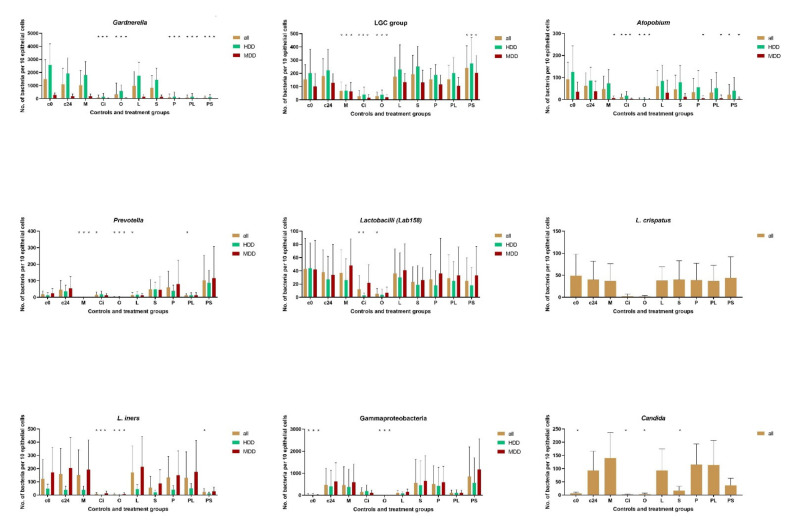
Mean ± SD concentrations of omnipresent *Gardnerella* and LGC bacteria surrounding 10 epithelial cells in the antimicrobial tests. The abbreviations are the same for all figures and tables in this manuscript: N -number of patients, c0—baseline controls, c2/c24—controls after 2 h and 24 h incubation, M—metronidazole, Ci—ciclopirox, O octenisept®, L—*Lactobacillus crispatus,* S—*Sacharomyces boulardii,* P—phagolysin, PL—phagolysine + *Lactobacillus crispatus*, PS—phagolysin + *Sacharomyces boulardii.* HDD—high density dysbiosis, MDD moderate density dysbiosis. “*” indicates statistical significance with *p* < 0.05 as assessed with one-way ANOVA with multiple comparisons against the c24 control.

**Figure 2 pathogens-11-00456-f002:**
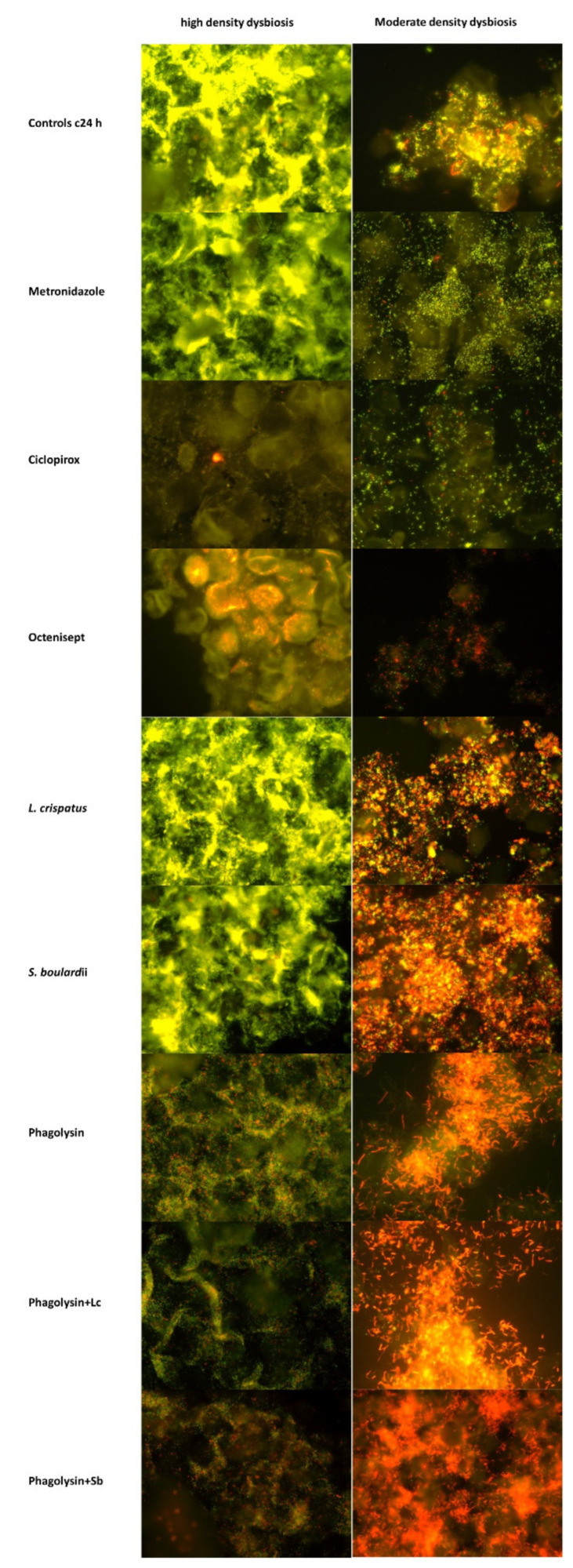
Multicolor FISH demonstrates the effects of antimicrobials on *Gardnerella* (Gard662 C3 orange fluorescence) and Low GC bacteria (LGC 35-Cy5 dark red fluorescence) at magnification ×400 in two BV patients representing high and moderate density dysbiosis. The changes in corpuscular integrity of bacteria in vaginal microbiome can be well seen. Susceptible bacteria either disappear or they lose their structure according to the grade/range of disintegration after the applications of specific antimicrobial drugs. Due to selective effects of antimicrobials on different species, the proportions of bacteria in the samples change dramatically. Low concentrated species previously outnumbered by dominant representatives and imperceptible become apparent. Resistant to antimicrobial species start to grow and become predominant groups within the in vitro cultured microbiome. This leads in multicolor FISH to dramatic changes in color of the predominant population. *Gardnerella* species (orange fluorescence) dominate the microbiome population in controls after 24 h incubation without antimicrobials. Because of the high numbers of *Gardnerella,* Low-GC bacteria cannot be seen well (red fluorescence). Gardnerella still is still dominant in metronidazole*, L. crispatus* and *Saccharomyces boulardii* (high density dysbiosis) exposed samples. *Gardnerella* is decimated in other test sets dismantling LGC bacteria, when those were not likewise lysed with octenisept or ciclopirox.

**Table 1 pathogens-11-00456-t001:** Mean concentrations of omnipresent *Gardnerella* and LGC bacteria surrounding 10 epithelial cells in the antimicrobial tests.

	*Gardnerella*			LGC Group		
	All	HDD	MDD	All	HDD	MDD
N=	38	20	18	38	20	18
c0	1494	2576	280	156	202	101
c2	1380			161		
c24	1097	1917	187	179	224	129
**M**	1032	1803	174	**67**	**70**	**63**
**Ci**	**93**	**153**	**26**	**28**	**40**	**16**
**O**	**331**	**587**	**47**	**29**	**39**	**19**
**L**	997	1756	154	175	229	134
**S**	816	1430	134	194	251	131
**P**	**97**	**161**	**26**	154	188	116
**PL**	**88**	**150**	**19**	156	202	105
**PS**	**63**	**120**	**1**	**242**	**275**	**204**
	k24/c,o,p,pl,ps *p* < 0.001	24/c,o,p,pl,ps *p* < 0.001	k24/c,o,p,pl,ps *p* < 0.05 p/ps *p* < 0.01	k24/m,c,o *p* < 0.001 k24/p/ps *p* = 0.01 k24/ps *p* = 0.03	k24/m *p* < 0.001 k24/p/ps *p* = 0.012 k24/ps *p* = 0.03	k24/m *p* < 0.02 k24/c,o *p* < 0.001 k24/ps *p* = 0.03 p/ps *p* = 0.01

SD values are not demonstrated since irrelevant for presentation. The abbreviations are the same for all tables in this manuscript: N -number of patients, c0—baseline controls, c2/c24—controls after 2 h and 24 h incubation, M—metronidazole, Ci—ciclopirox, O octenisept®, L—*Lactobacillus crispatus*, S—*Sacharomyces boulardii*, P—phagolysin, PL—phagolysine + *Lactobacillus crispatus*, PS—phagolysin + *Sacharomyces boulardii*. HDD -high density dysbiosis, MDD moderate density dysbiosis. Bold numbers within the table are the numbers significantly reduced as compared to 24 h controls.

**Table 2 pathogens-11-00456-t002:** Mean concentrations of bacteria detected only in some vaginal samples in the antimicrobial tests.

	*Atopobium*			Lactobacilli (Lab158)			*L.iners*			*L.crisp*	*Prevotella*			Gammaproteobacteria			*Candida*
	All	HDD	MDD	All	HDD	MDD	All	HDD	MDD	All	All	HDD	MDD	all	HDD	MDD	All
N=	24	16	8	25	15	10	14	5	9	5 (3)	24	13	11	15	5(2)	10 (4)	6(2)
	(64%)	(80%)	(44%)	(66%)	(75%)	(56%)	(37%)	(25%)	(50%	(13%)	(63%)	(65%)	(61%)	(39%)	(25/15%)	(56/28%)	(16%)
c0	92	125	36	43	44	42	123	49	172	49	17	13	24	**23**	**22**	**15**	**6.3**
c24	62	86	38	38	27	34	160	40	204	40	46	37	54	470	412	631	93
**M**	48	73	**7**	37	26	48	152	40	192	37	**0.6**	**0.7**	**0.5**	458	377	597	140
**Ci**	**10**	**17**	**3.3**	**12**	**2.8**	22	**8.4**	**1.2**	**13**	2.8	**16**	19	12	148	193	111	**2**
**O**	**2.6**	**3.6**	**1.4**	**5.3**	3.7	6.9	**4.9**	**2.8**	**6.8**	1.8	**1**	**1.5**	**0.8**	**9.6**	**9.2**	**9.9**	**3.6**
**L**	61	84	31	36	30	41	170	45	214	38	**13**	16	11	118	84	157	93
**S**	46	78	12	23	19	26	56	22	87	40	48	49	46	556	460	652	**17**
**P**	33	55	**5.8**	27	18	36	132	41	150	39	60	39	80	510	425	595	115
**PL**	32	52	**5.5**	29	25	33	131	50	176	37	**12**	13	12	115	115	116	114
**PS**	**22**	40	**3.2**	25	18	33	**23**	12	28	44	103	88	116	858	569	1168	37
	k24/ps *p* = 0.02 k24/c,o *p* < 0.001	k24/c,o *p* < 0.001	k24/m,c, o,l,s,p,pl, ps *p* < 0.05–0.02	K24/c,o *p* < 0.05	K24/c *p* < 0.05		K24/c,o *p* < 0.01 K24/ps *p* > 0.05	k24/c,o *p* < 0.05	k24/c,o *p* < 0.05	ns	k24/m *p* < 0.001 k24/c,o,l *p* < 0.05	k24/m *p* < 0.01 k24/o *p* < 0.05	k24/m *p* < 0.01 k24/o *p* < 0.05	k0/k24; k24/0 *p* < 0.01	k0/k24; k24/o *p* < 0.05	k0/k24; k24/0 *p* < 0.05	k0/k24; k24/c,o *p* = 0.01 k24/s,m *p* < 0.05

**Table 3 pathogens-11-00456-t003:** Susceptibility of vaginal microbiota to lysis as related to untreated controls.

	*Gardnerella*			LGC			*Atopobium*			Lactobacilli			*L.iners*			*L.crisp.*	*Prevotella*-*Bact.*			*Gammaproteobac.*			*Candida*
	All	HDD	MDD	All	HDD	MDD	All	HDD	MDD	All	MDD	HDD	All	HDD	MDD	All	All	MDD	HDD	All	HDD	MDD	All
M	**R**	**R**	**R**	**I**	**I**	**I**	**R**	**R**	**I**	**R**	**R**	**P**	**R**	**R**	**R**	**R**	**S**	**S**	**SS**	**R**	**R**	**R**	**P**
C	**S**	**S**	**I**	**I**	**I**	**I**	**I**	**I**	**S**	**I**	**I**	**I**	**S**	**S**	**S**	**S**	**I**	**I**	**I**	**I**	**I**	**I**	**S**
O	**I**	**I**	**I**	**I**	**I**	**I**	**S**	**S**	**S**	**I**	**I**	**I**	**S**	**S**	**S**	**S**	**S**	**S**	**S**	**S**	**S**	**S**	**S**
L	**R**	**R**	**R**	**R**	**R**	**R**	**R**	**R**	**R**	**R**	**R**	**R**	**R**	**R**	**R**	**R**	**I**	**I**	**I**	**I**	**I**	**I**	**R**
S	**I**	**I**	**I**	**R**	**R**	**R**	**I**	**R**	**I**	**I**	**I**	**R**	**I**	**I**	**I**	**R**	**R**	**R**	**R**	**R**	**R**	**R**	**I**
P	**S**	**S**	**I**	**R**	**R**	**R**	**I**	**I**	**I**	**I**	**I**	**R**	**R**	**R**	**I**	**R**	**R**	**R**	**P**	**R**	**R**	**R**	**R**
PL	**S**	**S**	**S**	**R**	**R**	**R**	**I**	**I**	**I**	**R**	**R**	**R**	**R**	**R**	**R**	**R**	**I**	**I**	**I**	**I**	**I**	**I**	**R**
PS	**S**	**S**	**SS**	**P**	**R**	**P**	**I**	**I**	**S**	**I**	**I**	**R**	**I**	**I**	**I**	**R**	**P**	**P**	**P**	**P**	**P**	**P**	**I**

R-resistant; S-sensitive; SS-highly sensitive; I-intermediate; P-promoting.

## Data Availability

The data of this publication were not presented previously. The work has not been published previously and is not under consideration for publication elsewhere. The publication is approved by all authors and tacitly or explicitly by the responsible authorities where the work was carried out. If accepted, it will not be published elsewhere in the same form, in English or in any other language, including electronically without the written consent of the copyright-holder.

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
