# Peer review of "Antimicrobial Susceptibility of Microbiota in Bacterial Vaginosis Using Fluorescence In Situ Hybridization"

_pathogens, 2022, doi:10.3390/pathogens11040456_

Round 1

Reviewer 1 Report

Figure 2 i very important. However, the explanation about is (below, lines 287, 288) is confusing. My suggestion to the authors is to provide a more extensive and more clear explanation. 

Author Response

Point by point response

Reviewer 1

English language and style are fine/minor spell check required

We have once more gone through the manuscript proving the spelling.

Figure 2 is very important. However, the explanation about is (below, lines 287, 288) is confusing. My suggestion to the authors is to provide a more extensive and more clear explanation. 

We have introduced the following (marked red)  explanation in the legend to figure 2

Figure 2. Multicolor FISH demonstrates the effects of antimicrobials on Gardnerella (Gard662 C3 orange fluorescence) and Low GC bacteria (LGC 35-Cy5 dark red fluorescence) at magnification x 400 in two BV patients representing high and moderate density dysbiosis.

The changes in corpuscular integrity of bacteria in vaginal microbiome can be well seen. Susceptible bacteria either disappear or they lose their structure according to the grade/range of disintegration after the applications of specific antimicrobial drugs. Due to selective effects of antimicrobials on different species, the proportions of bacteria in the samples change dramatically. Low concentrated species previously outnumbered by dominant representatives and imperceptible become apparent. Resistant to antimicrobial species start to grow and become predominant groups within the in vitro cultured microbiome. This leads in multicolor FISH to dramatic changes in color of the predominant population.

Gardnerella species (orange fluorescence) dominate the microbiome population in controls after 24 hours incubation without antimicrobials. Because of the high numbers of Gardnerella, Low-GC bacteria cannot be seen well (red fluorescence). Gardnerella is still dominant in metronidazole, L.crispatus and Saccharomyces boulardii (high density dysbiosis) exposed samples. Gardnerella is decimated in other test sets dismantling LGC bacteria, when these were not lysed with octenisept or ciclopirox.

Reviewer 2 Report

This is a very interesting paper with exciting and novel findings on an understanding of antimicrobial susceptibility of the vaginal microbiota. The experimental part was impeccably executed.

The authors showed that Lactobacillus crispatus, a probiotic strain, had the poorest bacteriolytical effects, followed by metronidazole. Both lysed Prevotella-Bacteroides, Enterobacteriaceae (by L.crispatus) or LGC, Atopobium, and Prevotella-Bacteroides (by metronidazole) groups of the microbiota, but had no effect on Gardnerella species. The authors also invstigated that Gardnerella cytolytic activity was highly pronounced and increased from octenisept to ciclopirox, phagolysin, phagolysin with L.crispatus, and phagolysin with S.boulardii, with the combination of phagolysin with S.boulardii being the best. Cycloproxyl and octenisept®, which are both universally active, repressed practically all microbial groups, including those that are considered helpful. Phagolysin exhibited no effect on Lactobacillus crispatus, a naturally occurring bacteria.

The paper is well organized and written, however, there are several concerns and suggestions:

Major concerns:

  1. The authors should provide more details and be accurate in the methodology section.

Minor concerns:

  1. The authors should increase the quality/size of figure 1 for better visualization.
  2. The authors should increase the quality/size of Figure 2 for better visualization. Some of the labeling/words are not visible.

Author Response

Point by point response

Reviewer 2

Extensive editing of English language and style required

Allow us to disagree. Two co-authors of the manuscript are full professors at American universities with more than 40 years of teaching history and hundreds of original publications. They thoroughly revised this manuscript from the linguistic and scientific point of view. Sending this highly specific manuscript to a language editor, unfamiliar with the specific clinical practice, would be an enormous detriment to the quality of the paper.

This is a very interesting paper with exciting and novel findings on an understanding of antimicrobial susceptibility of the vaginal microbiota. The experimental part was impeccably executed. The authors showed that Lactobacillus crispatus, a probiotic strain, had the poorest bacteriolytical effects, followed by metronidazole. Both lysed Prevotella-Bacteroides, Enterobacteriaceae (by L.crispatus) or LGC, Atopobium, and Prevotella-Bacteroides (by metronidazole) groups of the microbiota, but had no effect on Gardnerella species. The authors also invstigated that Gardnerella cytolytic activity was highly pronounced and increased from octenisept to ciclopirox, phagolysin, phagolysin with L.crispatus, and phagolysin with S.boulardii, with the combination of phagolysin with S.boulardii being the best. Cycloproxyl and octenisept®, which are both universally active, repressed practically all microbial groups, including those that are considered helpful. Phagolysin exhibited no effect on Lactobacillus crispatus, a naturally occurring bacteria.

Thank you for the comment. 

The paper is well organized and written, however, there are several concerns and suggestions:

Major concerns:

  1. The authors should provide more details and be accurate in the methodology section.

The FISH methodology is very extensive. Presenting it in detail, would turn the manuscript into a technical manual. All the methods used are well described in previous publications to which we reference. However, to ease the search for researches we added the following link to the methods sections, where the laboratory protocols are described more detailed.

The exact protocols with single steps and solutions are also presented at: http://www.swidsinski.de/zusatzdateien/fishmethode/fishmethode.htm

Minor concerns:

  1. The authors should increase the quality/size of figure 1 for better visualization.

We likewise feel that placing the Fig 1 horizontally, would increase its size by 50% and largely improve its visualization.

  1. The authors should increase the quality/size of Figure 2 for better visualization. Some of the labeling/words are not visible.

The original quality of the figures will be transferred to the journal upon acceptance of the manuscript. However, even in that case the figures concentrated on a single page cannot be fully appreciated because the large amount of visual information encoded in each. It would probably be a good idea to present the Figure 2 in two formats. As a print version as it is now, which is essential for understanding. In addition, as an online version in the original format, so that the reader could zoom single microphotographs to the size of the maximal resolution. We would be thankful to the reviewer if he could support this idea to the journal editors.

This manuscript is a resubmission of an earlier submission. The following is a list of the peer review reports and author responses from that submission.

Round 1

Reviewer 1 Report

1. in the work (text, tables, figures) wrong names of microorganisms are used, wrong nomenclature, e.g. replace "Gardnerella" with "Gardnerella vaginalis / Gardnerella spp." , "Candida" with "Candida spp." , 'Bacillalis "with" Bacillales "etc.

2. in microbiology, in the nomenclature of microorganisms, the names of types and orders are not used, e.g. "Firmicutes", "Bacillales", but rather generic and species names
3. cardinal error - the authors do not refer to any CLSI or EUCAST recommendations
4. CLSI and EUCAST do not define "sensible" and "SS" only "sensitive"
5. cardinal error - according to EUCAST, the term "intermediate" does not exist, only "sensitive, increased exposure" -
wrong nomenclature
6. Interpretation of results for metronidazole and ciclopirox should be based on the CLSI and / or EUCAST recommendations
on what basis were the microbial strains classified as sensitive, resistant and intermediate?
7. disordered and poor literature, sometimes with no references in the text, despite the author's earlier work
8. How were the microorganisms identified, because in the case of Gardnerella spp. The species name - Gardnerella vaginalis was not even given?
8. otherwise, the work is very interesting and up-to-date

Author Response

Dear Editors of the Pathogens

Thank you for evaluating the manuscript for publication in your journal.

We revised the manuscript according to reviewers’ suggestions and upload the revised files together with this point-by-point response.

The changes in the manuscript are marked red for better tracking.

Best wishes

Alexander Swidsinski

Point by point response:

Reviewer 1

 Comments and Suggestions for Authors

  1. in the work (text, tables, figures) wrong names of microorganisms are used, wrong nomenclature, e.g. replace "Gardnerella" with "Gardnerella vaginalis / Gardnerella spp." , "Candida" with "Candida spp." , 'Bacillalis "with" Bacillales "etc.
  2. in microbiology, in the nomenclature of microorganisms, the names of types and orders are not used, e.g. "Firmicutes", "Bacillales", but rather generic and species names

Resp:

In contrast to classic culture-based microbiology, bacteria detected by sequence based molecular-genetic methods are not available as living organisms for investigation. Many were never isolated or cultured in vivo as species, and the real diversity of the microorganisms covered by sequencing of, in this cases, 16-23 s RNA is much broader, than supposed from culture data (resp. to 2.). Molecular-genetic methods detect clusters, corresponding to known microorganism. Adding sp or ssp behind would suggest, that we are working with species, which will be incorrect. Using word cluster behind each name is however completely confusing and not usual in the scientific literature dedicated to FISH detection of microorganisms.

We corrected the 'Bacillalis "to" Bacillales as proposed

  1. cardinal error - the authors do not refer to any CLSI or EUCAST recommendations

Resp:

Not including reference is deliberate. CLSI or EUCAST recommendations are absolutely inapplicable for the presented approach for the following reasons:

EUCAST clinical breakpoints - breakpoints and guidance (January 6, 2021) are developed for the evaluation of microbial growth under the impact of antibiotics. We are investigating microbial lysis.

The CLSI or EUCAST breakpoints completely omit probiotics, phages lysins, antiseptics and topical antifungals, which constitute the main body of antimicrobial substances tested in this paper.

CLSI or EUCAS do not apply to topical therapy, which is solely referred to in our paper.

Specifically, CLSI or EUCAST do not contain breakpoints for any of the tested in this study antimicrobials applied as a topical therapy.

  1. CLSI and EUCAST do not define "sensible" and "SS" only "sensitive".

Resp:

The terminology of CLSI and EUCAST is indeed most appropriate for the assessment of growth suppression of the pure culture, which is more exactly expressed via breakpoints. We are measuring lysis, which cannot be technically expressed in MICs. We apologize for incorrectly using German word “sensible” in tables instead of “sensitive” used in the main body of the manuscript. We corrected the mistake in tables.  

  1. cardinal error - according to EUCAST, the term "intermediate" does not exist, only "sensitive, increased exposure" - wrong nomenclature

Resp:

The term intermediate was toppled 2019 by EUCAST after it was used for decades. The remediation is reasonable from the MIC point of view, since antibiotic concentrations by general application have different concentrations inside of the human body being for example highly concentrated in urine, bile, specific tissues etc. and well working here despite “intermediate” MICs in vitro. Using “intermediate“ was therefore inappropriate, “when there is a high likelihood of therapeutic success, because exposure to the  (orally or intravenous applied) agent is increased by adjusting the dosing regimen or by its concentration at the site of infection” ( sited from https://www.eucast.org/newsiandr).

The use of EUCAST terminology in our manuscript would imply that using vaginal antimicrobials somewhere else would be valuable. These is a completely wrong conclusion in context of our experiments.

  1. Interpretation of results for metronidazole and ciclopirox should be based on the CLSI and / or EUCAST recommendations on what basis were the microbial strains classified as sensitive, resistant and intermediate?

Resp:

Allow us to disagree, since the European Committee on Antimicrobial Susceptibility Testing (Breakpoint tables for interpretation of MICs and zone diameters) Version 11.0, valid from 2021-01-01 does not include recommendations for topical treatment with metronidazole against any of the tested microbial clusters.  The only available in this tables data are referring to MICs of the generally applied metronidazole.

  1. disordered and poor literature, sometimes with no references in the text, despite the author's earlier work.

Resp:

We are preparing a literature review of the topic. We do not feel, that overloading this original work with references will add to the manuscript.

  1. How were the microorganisms identified, because in the case of Gardnerella spp. The species name - Gardnerella vaginalis was not even given?

 Resp:       

The name of Gardnerella vaginalis was not used deliberately. Recently, at least 12 new Gardnerella species were described, all of which hybridized with the classic Gardnerella vaginalis FISH probe. From all we know, it is highly likely that the Gardnerella cluster is even bigger than that.

  1. otherwise, the work is very interesting and up-to-date

Resp:

Thank you for the comment.

Reviewer 2 Report

Abstract:

The authors should provide enough background information to communicate the study significance.

The adequate use of abbreviation should be revised (e.g. L17).

As presented, the abstract is difficult to follow. The experimental questions, methodology and results, and conclusions are difficult to follow.

Main text:

Numerous sentences require a reference to support the claim, fact, or idea. For example, L40, L43, L44, L72, etc. The authors should revise the whole document and include references to support the stated claims.

As presented, the Introduction is verbose and difficult to follow. For example, L62-69. Idea depicted in L67-69 is very confusing.

Lines 70-71 is confusing, and L70-76 requires references to support the stated claims.

Transitions and paragraph flow must be improved in the Introduction.

As presented, the logic behind experimental question is difficult to follow. In L80-85 it reads:

Although FISH cannot discriminate between living and dead microorganisms (microbes are fixed at the time of FISH), the disappearance of the intact microbial cells after incubation is an unquestionable sign of a lytic bactericidal activity and can be utilized for susceptibility assessment. In the present study, we quantified the bacteriolytic effects of six common antimicrobials reported to have clinical efficacy and representing different modes of action.

How is possible to evaluate the bacteriolytic effect if it is not possible to discriminate between live and dead bacteria ?

Methods:

The authors must provide evidence of the ethics approval and informed consent statements for this study.

The authors should provide evidence of the inhouse validation process for the Antimicrobial susceptibility test, Susceptibility test and FISH assays.

Results and conclusions

It is not clear how the use of a FISH assay could provide reliable information about the bacteriolytic effect. This notion should be validated through a series of independent experiments. Overall, the experimental design and results lack appropriate validation process. Also, these observations are not corroborated. As presented, this contribution seems to preliminary to be considered for publication.

Author Response

Point by point response:

Reviewer 2   Comments and Suggestions for Authors

Abstract:

The authors should provide enough background information to communicate the study significance.

Resp.:

We reshaped the methods part of the abstract according to reviewers’ proposal, however the journal regulations limit the abstract to maximum of 200 words. This leaves us no possibility to expand the abstract.

The adequate use of abbreviation should be revised (e.g. L17).

Resp.:

We changed BV to bacterial vaginosis in line 17

As presented, the abstract is difficult to follow. The experimental questions, methodology and results, and conclusions are difficult to follow.

Resp.:

We restricted and specified the abstract remaining in limit of 200 words.

Main text:

Numerous sentences require a reference to support the claim, fact, or idea. For example, L40, L43, L44, L72, etc. The authors should revise the whole document and include references to support the stated claims.

Resp.:

We added the reference numbers behind each sentence.

 As presented, the Introduction is verbose and difficult to follow. For example, L62-69. Idea depicted in L67-69 is very confusing.

Resp.:

We reshaped the lines.

Lines 70-71 is confusing, and L70-76 requires references to support the stated claims.

Resp.:

We added the exact specification of the Eswab in the text, which is at the same time the most exact reference ESwab™ 493C02 COPAN, confirming our statement.

 Transitions and paragraph flow must be improved in the Introduction.

Resp.:

Transitions and paragraphs were introduced by journal compilation software. They were mainly absent in the originally submitted word version. We will follow your recommendations by the proofs.

As presented, the logic behind experimental question is difficult to follow. In L80-85 it reads:

Although FISH cannot discriminate between living and dead microorganisms (microbes are fixed at the time of FISH), the disappearance of the intact microbial cells after incubation is an unquestionable sign of a lytic bactericidal activity and can be utilized for susceptibility assessment. In the present study, we quantified the bacteriolytic effects of six common antimicrobials reported to have clinical efficacy and representing different modes of action.

 How is possible to evaluate the bacteriolytic effect if it is not possible to discriminate between live and dead bacteria?

Resp.:

We changed the sentence adding to manuscript: Visual disappearance of previously documented and visualized bacteria indicates that they were lysed by the tested substances.

Methods:

The authors must provide evidence of the ethics approval and informed consent statements for this study.

Resp.:

The documents were provided for the journal. Confirming response is below:

“”Please be noted that we can confirm that we have received ethical document
for your study, but it is not provided alongside with your manuscript when we
send for peer-review. With regards to one reviewer's ethical concern, you
could explain that you have provided the relevant document to our editorial
office”

The authors should provide evidence of the inhouse validation process for the Antimicrobial susceptibility test, Susceptibility test and FISH assays.

Resp.:

Molecular-genetic laboratory for polymicrobial infections of the Charite university hospital Berlin has existed since 1995, conducted and published many FISH studies.

MDI Limbach GmbH Laboratory is the leading institution in Berlin and for 60 years routinely performed antibiotic testing for Berlin”s hospitals and practices. No changes to the manuscript.

Results and conclusions

It is not clear how the use of a FISH assay could provide reliable information about the bacteriolytic effect. This notion should be validated through a series of independent experiments. Overall, the experimental design and results lack appropriate validation process. Also, these observations are not corroborated. As presented, this contribution seems to preliminary to be considered for publication.

Resp.:

We acknowledge that this report does not answer all questions involved in this new methodology. However, this is the first effort made to approach susceptibility testing of a composite complex community of bacterial rather than each individual species involved. Additional studies are needed. This report serves as the first introductory study addressing the problem and supports several clinical observations.

We have added this statement to the final conclusion.